# *MiR-375* Regulation of LDHB Plays Distinct Roles in Polyomavirus-Positive and -Negative Merkel Cell Carcinoma

**DOI:** 10.3390/cancers10110443

**Published:** 2018-11-14

**Authors:** Satendra Kumar, Hong Xie, Patrick Scicluna, Linkiat Lee, Viveca Björnhagen, Anders Höög, Catharina Larsson, Weng-Onn Lui

**Affiliations:** 1Department of Oncology-Pathology, Karolinska Institutet, SE-17176 Stockholm, Sweden; satendra.kumar@ki.se (S.K.); xiehong@tmu.edu.cn (H.X.); patrick.scicluna@ki.se (P.S.); Linkiatlee@gmail.com (L.L.); anders.hoog@ki.se (A.H.); catharina.larsson@ki.se (C.L.); 2Cancer Center Karolinska, Karolinska University Hospital, SE-17176 Stockholm, Sweden; 3Tianjin Life Science Research Center and Department of Pathogen Biology, School of Basic Medical Sciences, Tianjin Medical University, Tianjin 300070, China; 4Department of Cell and Molecular Biology, Karolinska Institutet, SE-17165 Stockholm, Sweden; 5Department of Reconstructive Plastic Surgery, Karolinska University Hospital, SE-17176 Stockholm, Sweden; viveca.bjornhagen-safwenberg@sll.se; 6Department of Clinical Pathology and Cytology, Karolinska University Hospital, SE-17176 Stockholm, Sweden

**Keywords:** Merkel cell carcinoma, Merkel cell polyomavirus, *miR-375*, LDHB, cell growth

## Abstract

*MicroRNA-375* (*miR-375*) is deregulated in multiple tumor types and regulates important targets involved in tumorigenesis and metastasis. This miRNA is highly expressed in Merkel cell carcinoma (MCC) compared to normal skin and other non-MCC skin cancers, and its expression is high in Merkel cell polyomavirus (MCPyV)-positive (MCPyV+) and low in MCPyV-negative (MCPyV−) MCC tumors. In this study, we characterized the function and target of *miR-375* in MCPyV+ and MCPyV− MCC cell lines. Ectopic expression of *miR-375* in MCPyV− MCC cells resulted in decreased cell proliferation and migration, as well as increased cell apoptosis and cell cycle arrest. However, in MCPyV+ MCC cells, inhibition of *miR-375* expression reduced cell growth and induced apoptosis. Additionally, the expression of lactate dehydrogenase B (*LDHB*), a known target of *miR-375*, was inversely correlated with *miR-375*. Silencing of LDHB reduced cell growth in MCPyV− cell lines, while its silencing in MCPyV+ cell lines rescued the cell growth effect mediated by *miR-375* inhibition. Together, our results suggest dual roles of *miR-375* and LDHB in MCPyV and non-MCPyV-associated MCCs. We propose that LDHB could be a therapeutic target in MCC and different strategies should be applied in virus- and non-virus-associated MCCs.

## 1. Introduction

Merkel cell carcinoma (MCC) is a highly aggressive form of neuroendocrine cancer of the skin. The majority of cases are caused by the Merkel cell polyomavirus (MCPyV) [1] that was found monoclonally integrated into cancer cell genome with tumor-specific T antigen mutations of importance for MCC tumorigenesis [2,3,4,5,6,7,8,9,10,11,12]. Besides, ~20% of MCC tumors do not have detectable MCPyV, indicating different etiologies and clinical outcomes between MCPyV-positive (MCPyV+) and MCPyV-negative (MCPyV−) MCC tumors. In line with this distinction, we and others have reported different clinical and molecular features between MCPyV+ and MCPyV− MCCs [13,14,15,16,17,18]. In our previous study, we identified a subset of differentially expressed microRNAs (miRNAs) between MCPyV+ and MCPyV− MCC tumors [18]. Among them, *microRNA-375* (*miR-375*) expression was found to be significantly higher in the MCPyV+ than the MCPyV− MCC tumors. Importantly this miRNA is highly specific to MCC compared to non-MCC tumors and cell lines [19,20]; its expression is also higher in serum samples of MCC patients than tumor-free patients or healthy individuals, suggesting its potential use as a surrogate marker for tumor burden in MCC [19].

Deregulation of *miR-375* expression has been reported in multiple types of cancer. Typically, *miR-375* is under-expressed such as in gastric carcinoma [21,22], glioma [23], colon cancer [24,25], head and neck squamous cell carcinoma [26], pancreatic cancer [27], hepatocellular carcinoma [28], and squamous cervical cancer [29]. However, overexpression of *miR-375* has also been observed in medullary thyroid carcinoma [30], breast cancer [31], and prostate cancer [32]. 

Functionally, *miR-375* acts as a tumor suppressor that inhibits cell proliferation, migration, invasion, and tumor metastasis [21,28,29,33] by targeting several important oncogenes, including phosphoinositide-dependent protein kinase-1 (PDK1) [22,34], 14-3-3 protein zeta (14-3-3ζ) [22,35], Yes-associated protein (YAP) [36], astrocyte elevated gene-1 (AEG1) [28,37,38], lactate dehydrogenase B (LDHB) [39], and Janus kinase 2 (JAK2) [21]. On the other hand, an oncogenic role of *miR-375* has been demonstrated in prostate and breast cancers. In prostate cancer, *miR-375* is highly expressed in high-grade and metastatic tumors, and its over-expression increases cell growth [40], while in breast cancer, its inhibition led to decreased cell growth [31]. In MCC, ectopic expression of *miR-375* in MCPyV− MCC cell lines was found to promote neuroendocrine differentiation and exhibit tumor suppressor effects [41]. However, its functional role in MCPyV+ MCCs is yet to be determined. 

Among the *miR-375* targets, LDHB was found upregulated in MCC tumors compared to carcinoid tumors of the lung, based on a proteomic approach [42]. LDHB is a key enzyme that catalyzes the conversion of lactate to pyruvate and NAD+ to NADH (an oxidized and reduced form of nicotinamide adenine dinucleotide, respectively) and is known to play important roles in cancer cell growth and progression [43,44,45]. Similar to the dual roles of *miR-375*, LDHB has also been demonstrated to enhance or suppress tumorigenesis depending on the cellular context [43,44]. Although LDHB was detected in some MCC tumors [42], its functional role in MCC remains unknown. 

In this study, we investigated the relation between expression of *miR-375* and LDHB, and their functional roles in MCC. We observed that LDHB expression was inversely correlated with *miR-375* levels. Interestingly, LDHB was found to have distinct roles in MCPyV+ and MCPyV− MCC cells.

## 2. Results

### 2.1. miR-375 and LDHB Expression Levels Are Inversely Correlated in MCC

To evaluate whether LDHB could be a potential target of *miR-375* in MCC, we quantified *miR-375* and LDHB expressions in three MCPyV− (MCC13, MCC14/2, and MCC26) and MCPyV+ (MKL-1, MKL-2 and WaGa) MCC cell lines using RT-qPCR and Western blotting, respectively. *miR-375* levels were higher in all three MCPyV+ than the MCPyV− cell lines, whereas the LDHB levels were opposite (Figure 1A,B). To further establish the expression relationship between *miR-375* and LDHB, we compared *miR-375* and *LDHB* mRNA expressions in a series of 54 MCC tumor samples. Consistent with the observation in cell lines, *miR-375* was inversely correlated with *LDHB* mRNA levels (*p* < 0.0001, Spearman’s rank order correlation, Figure 1C).

### 2.2. miR-375 Regulates LDHB Expression in MCC Cells

We next assessed whether *miR-375* could regulate LDHB expression in MCC cell lines. We ectopically expressed *miR-375* using an expression plasmid (miR-375 OE) in the three MCPyV− cell lines and silenced *miR-375* using miRNA sponge (miR-375sp) in two MCPyV+ cell lines. Transfection with miR-375 OE increased *miR-375* levels in all three MCPyV− cell lines, while inhibition of *miR-375* (miR-375sp) reduced its levels in both WaGa and MKL-1 cells (Figure 2A). Furthermore, *miR-375* over-expression reduced and its inhibition increased LDHB mRNA and protein levels (Figure 2B,C). Taken together, these observations indicate that LDHB is a target of *miR-375* in MCC.

Given that *miR-375* is one of the MCC-specific miRNAs and its differential expression between MCPyV+ and MCPyV− MCC tumors, we sought to determine whether *miR-375* plays distinct roles in these two tumor entities.

### 2.3. Over-Expression of miR-375 Inhibits Cell Growth and Migration in MCPyV− MCC Cell Lines

To determine the effect of *miR-375* on tumor phenotypes, we ectopically expressed *miR-375* in the MCPyV− cell lines using miRNA mimic or expression plasmid and investigated its effect on cell growth, cell cycle, cell migration, and apoptosis. Using RT-qPCR, we validated increased *miR-375* levels in cells transfected with *miR-375* mimic or expression plasmid (miR-375 OE) (Figure 3A). Using WST-1 assay, we observed a decrease of cell growth after 48 h (MCC 14/2) or 72 h (MCC13 and MCC26) of transfection of *miR-375* mimic (Figure 3B). Similar to the effect with *miR-375* mimic, MCC14/2 cells stably transfected with miR-375 OE reduced cell growth after 48 h, as evaluated by WST-1 and trypan blue exclusion assays (Figure 3C). The results support similar effect in *miR-375* over-expressing cells using either *miR-375* mimic or expression plasmid.

Cell cycle analysis in MCC13 and MCC14/2 revealed that *miR-375* mimic-treated cells had a subtle increase (7–14%) of cells in G1 or G2 phases compared with the negative control cells, respectively (Figure 4A). Wound healing scratch assays revealed that ectopically expressed *miR-375* retarded wound closure compared with the negative control at 18 h or 27 h (Figure 4B). To determine the effect on apoptosis, we evaluated the cleavage products of Poly (ADP-ribose) polymerase (PARP) (an apoptotic marker) from cells over-expressing *miR-375* or negative control using Western blot analysis. As shown in Figure 4C, we observed that the 89-kDa cleavage product of PARP was increased in all three MCC cell lines over-expressing *miR-375* compared to miRNA mimic control, suggesting that *miR-375* expression induces cell apoptosis in MCPyV− MCC cells.

### 2.4. Inhibition of miR-375 Expression Reduces Cell Growth and Induces Apoptosis in MCPyV+ MCC Cells

We suppressed *miR-375* expression in two MCPyV+ MCC cell lines using miR-375sp (Figure 5A). Using WST-1 and trypan blue exclusion assays, we observed that suppression of *miR-375* led to decreased cell growth in both WaGa and MKL-1 cell lines (Figure 5B). To further examine whether the reduction of cell growth was due to apoptosis, we determined the apoptotic effect using Annexin V and caspase-3 activity assays. For Annexin V assay, we observed that suppression of *miR-375* increased the number of apoptotic cells by 13% (*p* = 0.016) compared to the vector control-transfected cells (Figure 5C). Concordantly, we also observed increased of caspase-3 activity upon suppression of *miR-375* (2.7-fold, *p* = 0.001; Figure 5D). Together, our results suggest that *miR-375* suppression inhibited cell growth via apoptosis in MCPyV+ MCC cells.

### 2.5. Silencing of LDHB Rescues Cell Growth Effect Mediated by miR-375 Suppression

To determine whether LDHB plays a role in *miR-375* regulation of cell growth, we compared cell growth in miR-375sp-transfected cells with and without silencing of LDHB using two different siRNAs (siLDHB #1 and siLDHB #2). In parallel, we also transfected cells with miR-375sp or vector control only. As shown in Figure 6A, cells transfected with miR-375sp only or together with siCTR had higher LDHB levels than the pcDNA3 vector control. Co-transfection of miR-375sp and siLDHB led to a decrease in LDHB levels compared to cells transfected with miR-375sp and siCTR. Consistently, we observed decreased cell growth upon inhibition of *miR-375*, in which the effect was rescued by silencing of LDHB (Figure 6B).

### 2.6. Silencing of LDHB Reduces Cell Growth in MCPyV− MCC Cells

In MCPyV− MCC cells, we observed higher LDHB levels (Figure 1B) and that over-expression of *miR-375* reduced cell growth (Figure 3B,C). We therefore asked whether silencing of LDHB could phenocopy the effect of *miR-375* down-regulation. Indeed, silencing of LDHB reduced cell growth and increased apoptosis (as indicated by increased cleaved PARP levels) (Figure 6C,D).

## 3. Discussion

MCC are generally divided into MCPyV+ and MCPyV− tumors, depending on their etiologies. While MCPyV+ tumors are the most common MCCs in US and Europe, the MCPyV− tumors are more common in Australia [17,46]. Numerous data indicate anatomical, genetical, and clinical differences between MCPyV+ and MCPyV− MCCs. Anatomically, MCPyV+ tumors are found more frequently on extremities, and MCPyV− tumors are more frequent in the head and neck [17,47]. Molecularly, MCPyV− MCCs harbor high mutation loads associated with ultra-violet (UV) signature, suggesting that UV exposure is the underlying etiology of MCPyV− MCCs [48]. On the contrary, MCPyV+ tumors have low mutation burdens, suggesting that the viral oncoproteins control key processes involved in MCC tumorigenesis [49]. Clinically, MCPyV− tumors are more aggressive, with increased risk of tumor progression and MCC-related death [17,47]. Additionally, MCPyV− and MCPyV+ MCCs may derive from different cell lineages [50]. All these observations support that MCPyV+ and MCPyV− MCCs are distinct tumor entities.

Given substantial differences between MCPyV+ and MCPyV− MCCs, we speculated that *miR-375* is functionally distinct between these two tumor types. Indeed, our results support that *miR-375* acts as a tumor suppressor in MCPyV− and function as an oncogene in MCPyV+ MCC cell lines. Consistent with our findings, low expression of *miR-375* and its tumor suppressor role has been observed in MCPyV− MCC cell lines [41]. In MCPyV+ cell lines, we observed that suppression of *miR-375* reduced cell growth and induced apoptosis, indicating that *miR-375* is important to maintain cell viability in virus-positive cells. *miR-375* is an MCC-specific miRNA and is highly expressed in MCPyV+ tumors and sera; it is thus not surprising that this miRNA plays pivotal roles in this tumor type.

Similar to MCC, *miR-375* is also expressed in other neuroendocrine or endocrine organs, including pancreas [51,52], pituitary [53], adrenal [54], thyroid [55,56], lung [57], and gastrointestinal tract [58]. This miRNA has been demonstrated to play important roles in regulating cell differentiation [41,57,58,59], hormone synthesis, and secretion [51,52,54]. In cancer, *miR-375* is generally downregulated and functions as a suppressor of cell growth, invasion, and migration in multiple tumor types [60]. Additionally, *miR-375* can regulate several cancer pathways, including Hippo, PI3K-Akt, Wnt, and Notch [60]. In MCPyV− MCC, it was demonstrated that *miR-375* could repress multiple targets of the Notch signaling that lead to suppression of cell viability, migration, and invasion [41]. 

It has also been shown that *miR-375* can directly repress the key glycolytic enzyme LDHB [39]. Given that MCPyV small T-antigen can promote glycolysis [61], we speculated that *miR-375* regulation of LDHB might be important in MCC tumorigenesis. Here, we demonstrated that LDHB mRNA and protein levels were reduced following over-expression of *miR-375* and increased after suppression of *miR-375*, supporting that *LDHB* is a target of *miR-375* in MCC. Functionally, we showed that silencing of LDHB could phenocopy the anti-survival effect of *miR-375* over-expression in MCPyV− MCC cell lines, indicating its oncogenic role. The results are consistent with previous studies supporting that LDHB promotes tumor development and progression [43]. However, in MCPyV+ MCC cell lines, silencing of LDHB could rescue the cell growth inhibition effect mediated by *miR-375* suppression, suggesting its role as a suppressor in MCPyV+ MCC. Similarly, reduced LDHB expression levels have also been observed in several cancer types, such as prostate cancer [44] and pancreatic cancer [62]. One common observation between these tumor types and MCPyV small T-antigen-transfected cells is their glycolytic phenotype. One possible explanation for the differential role of LDHB in MCPyV+ and MCPyV− MCC cell lines is that MCPyV+ cell lines rely on aerobic glycolysis, which requires continuous generation of NAD+ from LDHB suppression, while the oxidative cancer cells largely rely on LDHB activity to generate substrates for the Krebs cycle that fuels cellular activities. It is thus tempting to speculate that cellular metabolisms in MCPyV+ and MCPyV− MCC cells are different from one another; MCPyV+ cells are likely glycolytic and MCPyV− cells are oxidative. Given that the MCPyV small T-antigen can promote a pro-glycolytic phenotype, the question arises whether the viral oncoprotein could change the cellular metabolism of the cells that converts LDHB from its oncogenic role to tumor suppressor. Alternatively, the differential roles observed could be due to different cellular contexts rather than an effect of the virus itself. Further investigations are warranted to fully understand cellular metabolism differences between these two groups and whether MCPyV oncoproteins could change cellular metabolism of the cells or the function of LDHB.

## 4. Materials and Methods 

### 4.1. Cell Lines

Six MCC cell lines were included in this study. MCC13, MCC14/2, and MCC26 are MCPyV− cell lines, which were purchased from CellBank Australia (Westmead, Australia). WaGa, MKL-1, and MKL-2 are MCPyV+ cell lines, which were kindly provided by Drs. Jürgen C. Becker (Medical University of Graz, Graz, Austria), Nancy L. Krett (Northwestern University, Chicago, IL, USA), and Roland Houben (University Hospital Würzburg, Würzburg, Germany), respectively. All MCC cells were grown in RPMI-1640 medium supplemented with 10% (WaGa, MKL-1, and MKL-2) or 15% (MCC13, MCC14/2, and MCC26) fetal bovine serum at 37 °C with 5% CO_2_. The authenticity of the cell lines was verified by short tandem repeat (STR) profiling in our recent study [63].

### 4.2. MCC Tumor Samples

Twenty-six formalin-fixed paraffin-embedded (FFPE) and 28 frozen tumor samples were collected from the Karolinska University Hospital and Stockholm South General Hospital (Stockholm, Sweden). All samples had been included in our previous studies [18,63]. The study was approved by the Ethics Committee of Karolinska Institutet (2010/1092-31/3), and the use of archival materials was approved by the Karolinska University Hospital Biobank (BbK-00557). All materials were coded. The materials were obtained with written informed consent, except those samples collected prior to 2010, which at that time were covered by a general application of endocrine tumor collection approved by the ethic committee board of the Karolinska Institutet (Dnr. 91:86), and oral informed consent was applied.

### 4.3. RNA Extraction

Total RNA was extracted using mirVana miRNA isolation kit (Applied Biosystem/Ambion, Austin, TX, USA) and the concentrations were measured with the NanoDrop ND-1000 spectrophotometer (NanoDrop Technologies, Wilmington, DE, USA) and stored at −80 °C for further use.

### 4.4. TaqMan Reverse Transcription-Quantitative PCR (RT-qPCR)

Reverse transcription-quantitative polymerase chain reaction (RT-qPCR) was used to quantify *miR-375* and *LDHB* expressions using the StepOnePlus™ Real-Time PCR system (Life Technologies, Carlsbad, CA, USA). Predesigned TaqMan assays for mature *miR-375* (ID_000564), *RNU6B* (ID_001093), *LDHB* (Hs00929956_m1) and *GAPDH* (Hs99999905_m1) were purchased from Applied Biosystems. For mature *miR-375* and *RNU6B*, cDNA was synthesized from 120 ng total RNAs using TaqMan MicroRNA Reverse Transcription Kit (cat. no. 4366597; Applied Biosystems). For mRNAs, 100 ng total RNAs was used for cDNA synthesis using High Capacity cDNA Reverse Transcription kit (cat. no. 4368814; Applied Biosystems). All reactions were performed in triplicate. The relative expression levels of mature *miR-375* were normalized to *RNU6B*, while the *LDHB* expressions were normalized to *GAPDH*. The quantification of *miR-375* and *RNU6B* in 26 samples was previously analyzed [18], while the remaining samples were analyzed in this study.

### 4.5. Transfection Experiments

For over-expression of *miR-375*, 3 × 10^5^ cells of MCC13, MCC14/2, and MCC26 were transfected with 10 nM of mirVana *miR-375* mimic (MC10327, Ambion) or mirVana miRNA mimic Negative Control#1 (NC, AM17110; Ambion) using Lipofectamine^®^ RNAiMAX Reagent (Invitrogen, Carlsbad, CA, USA) or 1.5 μg of plasmid DNA (miR-375 OE or pcDNA3) using Lipofectamine 2000 (Invitrogen). Stable miR-375 OE-transfected cells were established by selection with G418 (1 mg/mL; Invitrogen) for at least four weeks. For inhibition of *miR-375*, 2 µg of miR-375sp, or pcDNA3 plasmid DNA was transfected into 4 × 10^6^ WaGa and MKL-1 cells using the Amaxa Cell Line Nucleofector kit V (program D-24 and A-24, respectively; Lonza, Basel, Switzerland). *miR-375* expression (miR-375 OE) and sponge (miR-375sp) vectors were generated in our previous studies [63].

Co-transfection of miR-375sp and siLDHB #1 (100 nM; SI03032589, Qiagen, Hilden, Germany), siLDHB #2 (100 nM; SI03052182, Qiagen) or control siRNA (100 nM; siCTR, SI03052182; Qiagen) were performed in 4 × 10^6^ of WaGa and MKL-1 cells using the same nucleofection protocol. For silencing of LDHB in MCPyV− cell lines, 3 × 10^5^ cells of MCC13, MCC14/2, and MCC26 were transfected with 10 nM of siLDHB #1, siLDHB #2, or siCTR using Lipofectamine 2000. 

### 4.6. WST-1 Cell Viability Assay

Cell proliferation was measured by using WST-1 (cat. no. 11644807001; Roche Applied Science, Mannheim, Germany) colorimetric assay. At different time points (24, 48, 72, or 96 h post-transfection), 10 μL of WST-1 reagent was added and incubated for 3 h (MCC13, MCC14/2, and MCC26), 2 h (WaGa), or 4 h (MKL-1) at 37 °C. Absorbance was determined at wavelengths 450 nm (measurement) and 650 nm (reference) using a VERSA max microplate reader (Molecular Devices, Sunnyvale, CA, USA). Each experimental group consisted of five or eight replicates for each time point and repeated three times independently.

### 4.7. Trypan Blue Exclusion Assay

Cells were stained with 0.4% trypan blue stain (Invitrogen) and analyzed using the TC10^TM^ automated cell counter (Bio-Rad, Hercules, CA, USA). Total live cells in the miR-375 OE, miR-375sp, and siLDHB-transfected cells were compared to their respective controls.

### 4.8. Cell Cycle Analysis

At 72 h after transfection, 1 × 10^6^ cells were washed with PBS and fixed in cold 50% ethanol for 1 h. After washing with PBS and treating with RNase A (0.2 mg/mL; R6513, Sigma-Aldrich, St. Louis, MO, USA) for 1 h at 37 °C, the cells were then stained with 10 μL propidium iodide (1 mg/mL; P4170, Sigma-Aldrich) and kept on ice in the dark. Cell cycle analysis was performed using flow cytometry (Cytomics FC 500; Beckman Coulter, Brea, CA, USA) and FlowJo software version 7.6.2 (Tree Star Inc., Ashland, OR, USA). All experiments were performed independently in triplicate.

### 4.9. Wound Healing Scratch Assay

After 48 h of transfection, a scratch wound was made on the confluent monolayer cells of each treatment group and cultured in low serum (2% FBS) medium. The scratch was imaged in real-time using IncuCyte S3 (Essen BioScience, Ann Arbor, MI, USA). Image J software version 1.43u (http://rsbweb.nih.gov/ij/) was used to process all images for quantification purposes. The wound closure (cell migration) was calculated by fraction of wound at the given time to the wound area at 0 h and normalized to viable cell number of transfected cells plated in parallel. Three independent replicates were included in each experimental group.

### 4.10. Apoptosis Assays

Cell apoptosis was evaluated in WaGa cells after 72 h of transfection with miR-375sp or pcDNA3 using Annexin V FITC Apoptosis kit (cat. no. 640905; BioLegend, San Diego, CA, USA) and Caspase-3 colorimetric assay (#K106; BioVision, Mountain View, CA, USA). All experimental conditions were performed according to the manufacturer’s instructions. The Annexin V and propidium iodide-stained cells were analyzed by NovoCyte flow cytometer (ACEA Biosciences, San Diego, CA, USA), and the caspase-3 cleavage products were measured at wavelength 405 nm using a VERSAmax microplate reader (Molecular Devices). All experiments were replicated three times independently.

### 4.11. Western Blot Analysis

Cells were harvested and lysed using NP-40 lysis buffer (FNN0021; Life Technologies), supplemented with 1 mM of phenylmethanesulfonyl fluoride (PMSF, Sigma-Aldrich) and protease inhibitor (complete protease inhibitor cocktail; Roche Diagnostics GmbH). Protein concentrations were measured using the Pierce^TM^ BCA Protein assay kit (Thermo Fisher scientific, Inc., Waltham, MA, USA). Twenty-five micrograms protein lysate were run in 4–12% NuPAGE SDS or 12% Bis-Tris gels (Invitrogen) and transferred to nitrocellulose membranes. Western blot membranes were incubated with LDHB (1:1000; A7625; ABclonal, Woburn, MA, USA), PARP (1:1000; #556362; BD Biosciences, Franklin Lakes, NJ, USA) and cleaved PARP (1:1000; ab32064; Abcam, Cambridge, UK) antibodies. GAPDH (1:10,000, sc-47724; Santa Cruz Biotechnology Inc., Santa Cruz, CA, USA or 1:5000, #5174 Cell Signaling technology, Danvers, MA, USA) was used for normalization. Signals were detected by LAS-1000 Image Analyzer (Fujifilm, Tokyo, Japan) and quantified by Image Gauge version 4.0 (Fujifilm).

### 4.12. Statistical Analysis

All analyses were performed using IBM SPSS Statistics version 24.0 (IBM Corp., Armonk, NY, USA) or MS Office Excel 2007. Paired Student’s *t*-test was performed to analyze transfection experiments. Spearman’s rank order correlation was used to evaluate correlation between *miR-375* and *LDHB* expressions. All analyses were 2-tailed, and *p*-values < 0.05 were regarded as significant.

## 5. Conclusions

We demonstrate distinct functional roles of *miR-375* and LDHB in MCPyV+ and MCPyV− MCCs. Targeting LDHB could be a novel therapy for MCC.

## Figures and Tables

**Figure 1 cancers-10-00443-f001:**
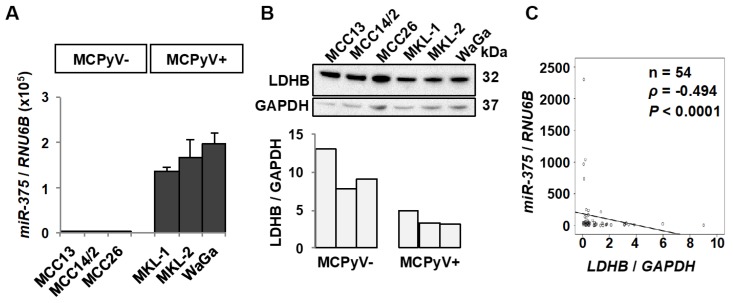
LDHB is inversely correlated with *miR-375* in MCC tumors and cell lines. (**A**) *miR-375* levels were quantified in six MCC cell lines using RT-qPCR. *RNU6B* was used for normalization purpose. Error bars represent SEM of four independent RNA isolations and RT-qPCR measurements. (**B**) Western blot analysis of LDHB protein expression in MCC cell lines. Western blot images of the LDHB and GAPDH levels are shown in the upper panel, and the quantifications of the LDHB levels are presented in the graph. The LDHB levels were normalized to GAPDH. (**C**) *miR-375* and *LDHB* mRNA levels were measured in 54 MCC tumor samples using RT-qPCR. The expression correlation between *miR-375* and *LDHB* mRNA was assessed by Spearman’s rank order correlation.

**Figure 2 cancers-10-00443-f002:**
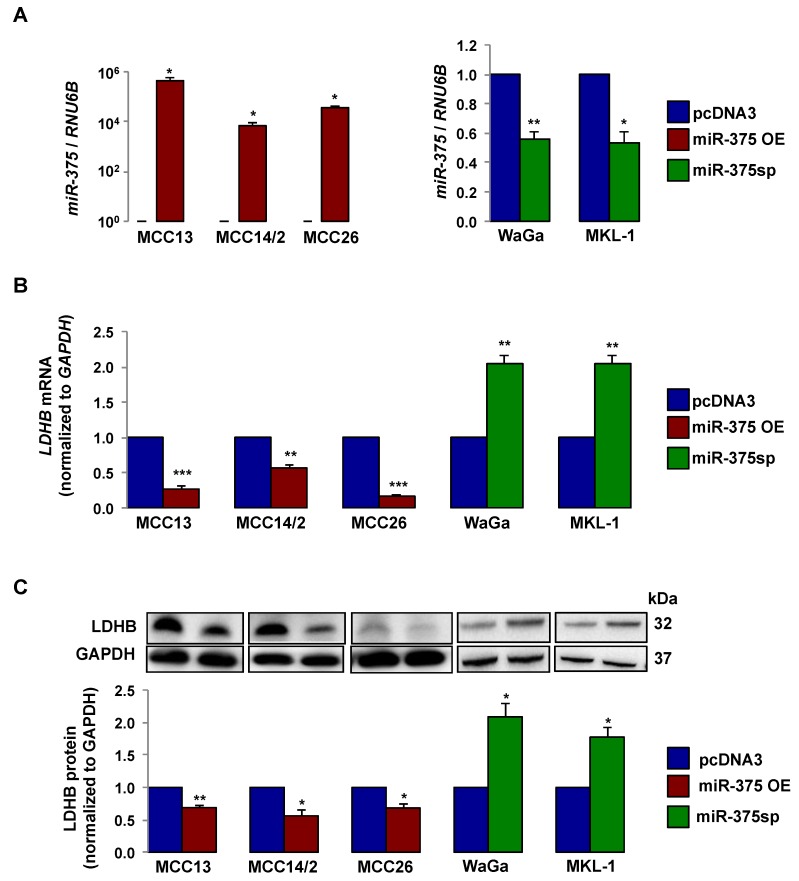
*miR-375* regulates LDHB mRNA and protein levels. (**A**) RT-qPCR analysis of *miR-375* levels in MCPyV− MCC cell lines (MCC13, MCC14/2, and MCC26) transfected with *miR-375* expression plasmid (miR-375 OE) or vector control (pcDNA3) after 48 h of transfection and in MCPyV+ MCC cell lines (WaGa and MKL-1) transfected with *miR-375* sponge (miR-375sp) or vector control (pcDNA3) after 72 h of transfection. The relative expression of *miR-375* was normalized to *RNU6B*. (**B**) Quantification of *LDHB* mRNA expressions in cells with over-expression or inhibition of *miR-375* in MCC cell lines by RT-qPCR. *LDHB* levels were normalized to *GAPDH* mRNA. (**C**) Western blot analysis of LDHB protein levels in cells with over-expression or inhibition of *miR-375*. Representative Western blot images are shown in the upper panel, and the quantifications of LDHB levels are presented in the graph below. Error bars represent SEM (n = 3). * *p* < 0.05, ** *p* < 0.01, and *** *p* < 0.001 by paired Student’s *t*-test.

**Figure 3 cancers-10-00443-f003:**
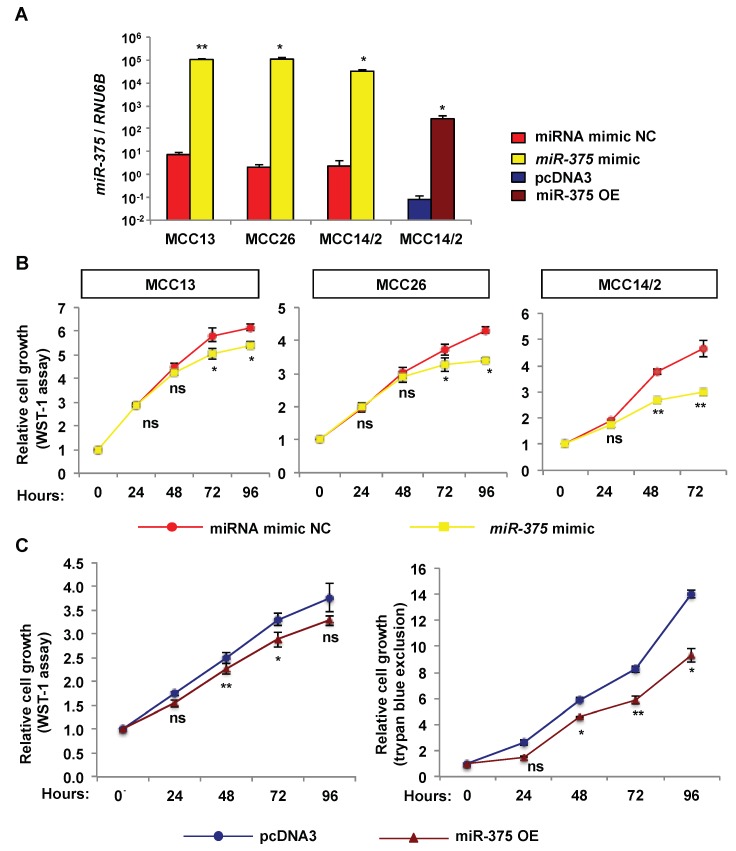
Over-expression of *miR-375* reduces cell growth in MCPyV− MCC cell lines. MCPyV− MCC cells were transfected with *miR-375* mimic or miR-375 OE and their respective negative controls. (**A**) Quantification of *miR-375* in the transfected cells after 48 h of transfection or stable miR-375 OE-transfected cells of MCC14/2 by RT-qPCR. The relative expression of *miR-375* was normalized to *RNU6B* and compared to miRNA mimic negative control (NC) or vector control (pcDNA3). (**B**) Evaluation of cell growth in cells transfected with *miR-375* mimic and miRNA mimic NC at different time points using WST-1 assay. (**C**) Cell growth in stable miR-375 OE-transfected cell line was evaluated at different time points using WST-1 and trypan blue exclusion assays. Error bars are SEM from three independent experiments. * *p* < 0.05 and ** *p* < 0.01 by paired Student’s *t*-test. ns = not significant.

**Figure 4 cancers-10-00443-f004:**
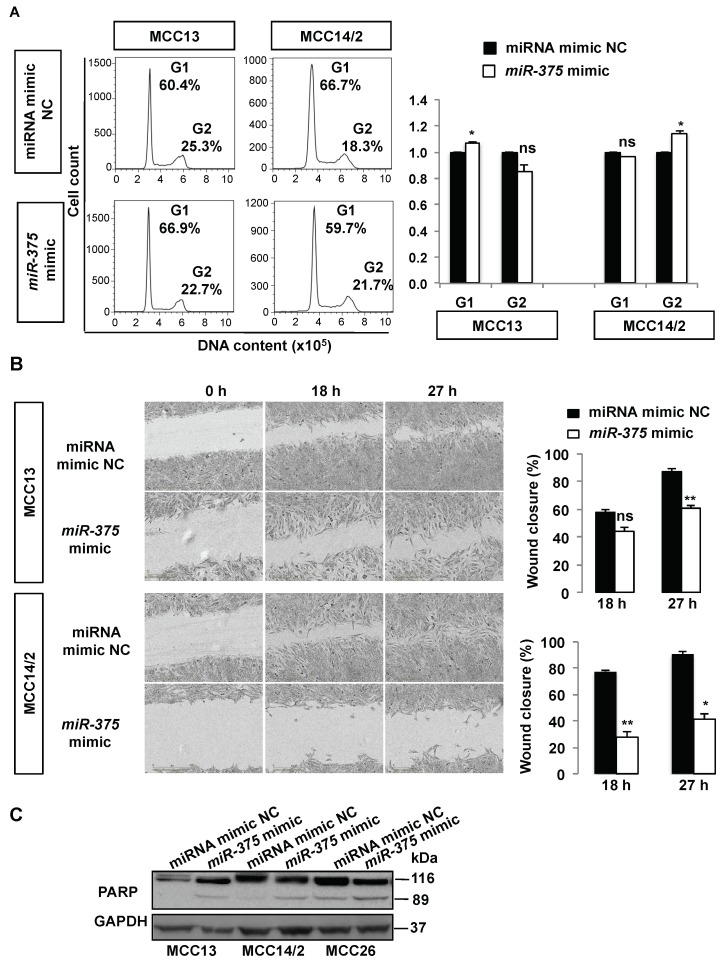
Over-expression of *miR-375* reduces cell migration and induces cell cycle arrest and apoptosis in MCPyV− MCC cell lines. MCC13 and MCC14/2 cells were transfected with *miR-375* mimic or miRNA mimic NC for 48 h. (**A**) Cell cycle analysis was evaluated in transfected cells by propidium iodide staining and flow cytometric analysis. *Left* panel: representative histograms illustrating the percentage of cells at G1 and G2 phases in cells with and without over-expression of *miR-375*. *Right* panel: Fractions of cells at G1 and G2 phases were calculated from the histograms of three independent experiments. (**B**) Cell migration was evaluated using wound-healing assay. *Left* panel: representative images of wound closure at 18 h and 27 h time points. *Right* panel: the wound closure was calculated based on the difference between wound gap at 18 h or 27 h and 0 h time point and normalized to viable cell number of transfected cells plated in parallel. (**C**) Apoptosis was evaluated after 48 h in MCC cells transfected with *miR-375* mimic or NC using Western blot analysis of anti-PARP (BD Pharmingen), which recognizes the full-length (116 kDa) and apoptosis-associated cleaved (89 kDa) forms. GAPDH was used as a loading control. * *p* < 0.05 and ** *p* < 0.01 by paired Student’s *t*-test. ns = not significant.

**Figure 5 cancers-10-00443-f005:**
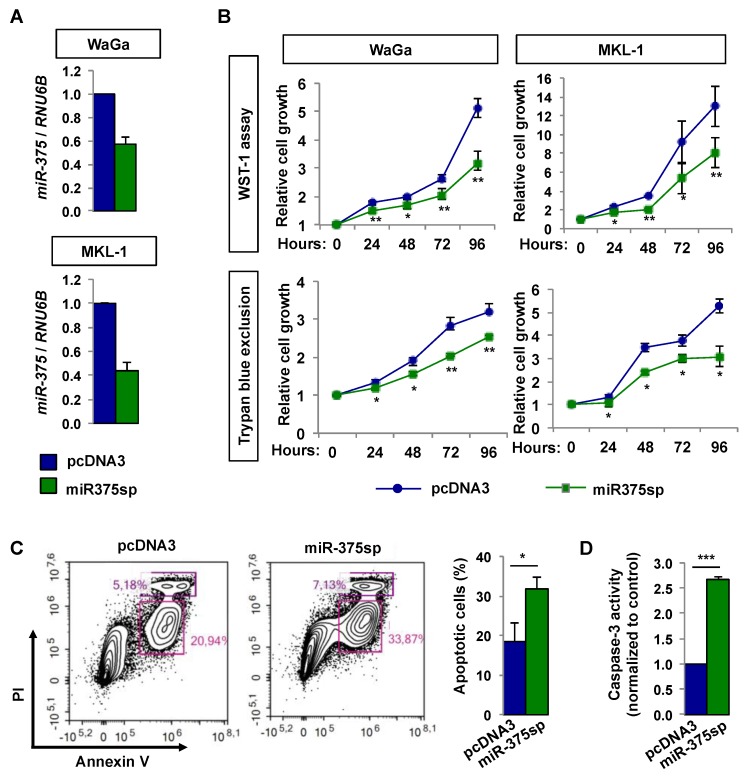
Inhibition of *miR-375* reduces cell growth and induces apoptosis in MCPyV+ MCC cell lines. WaGa and MKL-1 cells were transfected with miR-375sp or vector control (pcDNA3) for 72 h. (**A**) RT-qPCR analysis of *miR-375* levels in cells transfected with miR-375sp or pcDNA3. The *miR-375* expressions were normalized to *RNU6B*. (**B**) Effect of *miR-375* inhibition on cell growth was evaluated at different time points using WST-1 and trypan blue exclusion assays. (**C**) Representative flow cytometric images of WaGa cells co-stained with Annexin V-FITC and propidium iodide (PI) upon inhibition of *miR-375*. The apoptotic cells (Annexin V^+^/PI^−^) and the necrotic cells (Annexin V^+^/PI^+^) are represented in the lower and upper boxes, respectively. Quantification of the apoptotic cells is shown on the right panel. (**D**) Caspase-3 activity was quantified in WaGa cell lysates with and without inhibition of *miR-375*. Data are means ± SEM of three independent experiments. * *p* < 0.05, ** *p* < 0.01, and *** *p* = 0.001 by paired Student’s *t*-test. ns = not significant.

**Figure 6 cancers-10-00443-f006:**
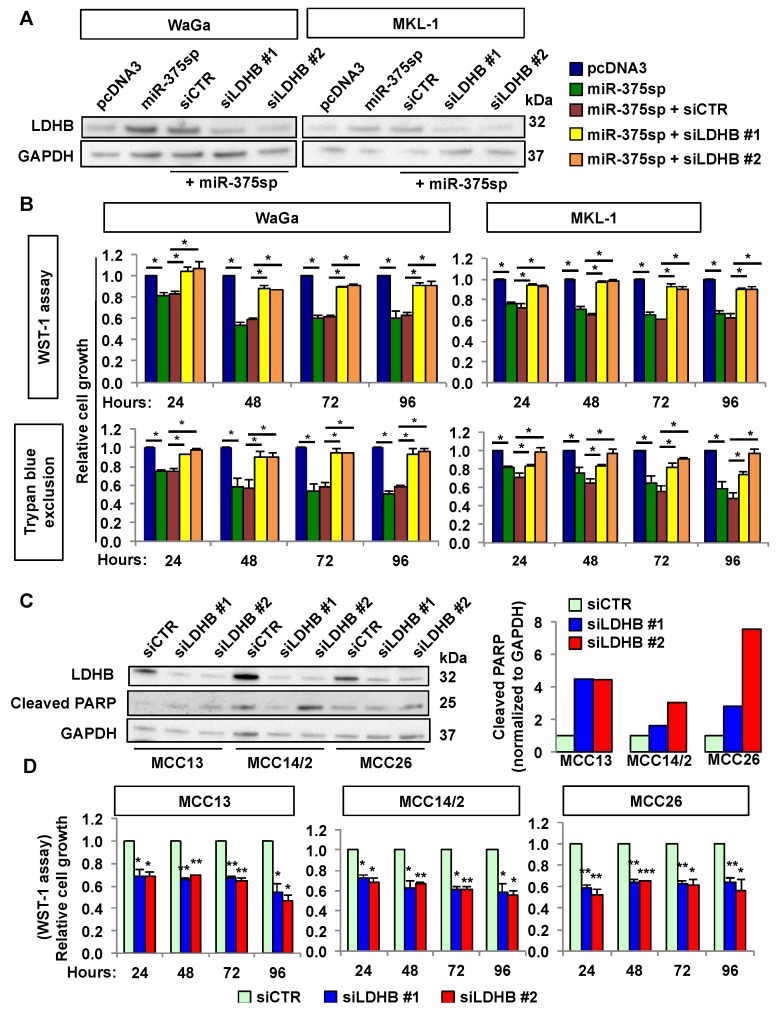
LDHB plays dual roles in MCPyV+ and MCPyV− MCC cells. (**A**,**B**) MCPyV+ MCC cell lines were transfected with miR-375sp or pcDNA3, or co-transfected with miR-375sp together with siCTR, siLDHB #1, or siLDHB #2. (**A**) Western blot analysis of LDHB in the transfected cells after 72 h of transfection. GAPDH was used as a loading control. (**B**) The effect on cell growth was evaluated at different time points using WST-1 and trypan blue exclusion assays. (**C**,**D**) MCPyV− MCC cell lines were transfected with siCTR, siLDHB #1, or siLDHB #2 for 48 h. (**C**) Western blot analysis of the effect of LDHB silencing on LDHB protein level and cleaved PARP. The specific 25 kDa cleaved form of PARP was detected using anti-cleaved PARP antibody (Abcam). Quantification of the cleaved PARP levels is presented on the right panel. (**D**) Effect of LDHB silencing on cell growth was evaluated using WST-1 assay. Mean ± SEM (n = 3). * *p* < 0.05, ** *p* < 0.01, and *** *p* < 0.001 by paired Student’s *t*-test.

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
