# Peer review of "MiR-375 Regulation of LDHB Plays Distinct Roles in Polyomavirus-Positive and -Negative Merkel Cell Carcinoma"

_cancers, 2018, doi:10.3390/cancers10110443_

Round 1

Reviewer 1 Report

The authors tried to clarify the role of miR-375 in two types of merkel cell carcinoma cells according to polyomavirus positiviity. The results indicated that miR-375 displayed a distinct role in these two types of merkel cell carcinoma cells with the same target, LDHB. Although the results seems to be clear, but more discussions should be made. The questions are listed:

1. It is not clear for the polyomavirus positivity in clinical samples (Fig. 1C). Could it possible to correlate with polyomavirus status among these samples?    

2. The metabolism analysis should be provided in these two types of merkel cell carcinoma cells to further support their conclusion about MCV- cells are oxidative and MCV+ cells are glycolytic.

3. The symbols of Fig. 6 should be changed to colorization to be better distinguished by readers.

4. The effect of miR-375 in cell migration of MCV- cells should be adjusted with cell survival.

5. The authors should discuss if any MCV proteins may change the functions of host cellular proteins such as LDHB from oncogenic to tumor suppression. I think that will be the most important/novel finding in this study. The authors should work hard on such discussions. 

Author Response

Point 1: “It is not clear for the polyomavirus positivity in clinical samples (Fig. 1C). Could it possible to correlate with polyomavirus status among these samples?”

Response: In this cohort, we included a total of 26 formalin-fixed paraffin-embedded (FFPE) and 28 frozen tumor samples, in which the virus status is available for all 26 FFPE samples detected by PCR and immunostaining, while the virus status for the frozen samples were detected by Western blotting. Among the 26 FFPE samples, 19 were MCPyV+ and 7 were MCPyV-. All frozen tumors were MCPyV+. Given the LDHB mRNA in FFPE and frozen tumors are not comparable, we could not combine both materials for comparison with MCPyV status. Based on the small cohort of FFPE samples, we did not find significant correlation between MCPyV status and LDHB levels. Given the low number of MCPyV- samples in this cohort, we propose not to include the analysis in this study.

Point 2: “The metabolism analysis should be provided in these two types of merkel cell carcinoma cells to further support their conclusion about MCV- cells are oxidative and MCV+ cells are glycolytic.”

Response:  We made a speculation that cellular metabolism could be different between MCPyV- and MCPyV+ cells. We agree that metabolic analysis is needed to support the statement, which will be included for future study. To clarify that the statement is not a conclusive remark, we revised the Conclusion as follows: We demonstrate distinct functional roles of miR-375 and LDHB in MCPyV+ and MCPyV- MCCs. Targeting LDHB could be a novel therapy for MCC.  

Point 3: “The symbols of Fig. 6 should be changed to colorization to be better distinguished by readers.”

Response: We fully agree with the suggestion from the reviewer. We have now revised Fig. 6 and show all graphs with color-coding. Furthermore, to improve the clarity of graphs presented throughout in the manuscript and considering the comments from Reviewer #2 we have similarly changed the graphs to color-coding in Figures 2, 3 and 5.

Point 4: “The effect of miR-375 in cell migration of MCV- cells should be adjusted with cell survival.”

Response: In our previous submission, we counted the number of viable cells before transfection and not before or after cell migration assay. We agree that reduced cell growth could have influenced the cell migration effect observed. We therefore repeated the wound healing scratch assay experiments with low serum concentrations in cell medium to suppress cell proliferation and normalized the cell migration effect to the number of viable cells. Due to low serum conditions, the cells were growing slower than our previous experiments with normal serum conditions, we therefore evaluated the cell migration effect at 18 h and 27 h time points. The new experiments are described in the Methods (page 10 line 429-433) and the results are presented in the revised Fig. 4B. 

Point 5: “ The authors should discuss if any MCV proteins may change the functions of host cellular proteins such as LDHB from oncogenic to tumor suppression. I think that will be the most important/novel finding in this study. The authors should work hard on such discussions. ”

Response: As described in the Discussion, we provide an explanation for the differential roles of LDHB in MCPyV+ and MCPyV- MCC cell lines. The MCPyV small T-antigen is known to promote pro-glycolytic metabolic changes, which requires continuous generation of NAD+ that benefits from LDHB suppression. Alternatively, MCPyV+ and MCPyV- MCC cells are likely to be derived from different cell types and it has been demonstrated that LDHB can have a dual role as an oncogene or tumor suppressor, depending on cellular context. The Discussion has been extended (line 292-343).  

Reviewer 2 Report

Expression levels of microRNA-375 (miR-375) are higher in Merkel cell polyomavirus-positive Merkel cell carcinoma cell lines and tumors than in virus-negative tumor cell lines and tumors. As Lactate dehydrogenase B (LDHB) mRNA is a target for miR-375, the authors found an inverse correlation between miR-375 and LDHB levels in virus-positive and –negative MCC cell lines. Overexpressing miR-375 in virus-negative MCC cell lines decreased proliferation and migration, but increased apoptosis and cell cycle arrest. siRNA-mediated depletion of LDHB  reduced cell growth in MCPyV-negative cell lines. In virus-positive MCC cell lines, the authors found that inhibition of miRNA-375 expression suppressed cell growth and stimulated apoptosis, while silencing LDHB expression rescued cell growth effect mediated by miR-375 suppression. The authors suggest that LDHB could be a therapeutic target in both MCPyV-positive and –negative MCCs, but different strategies are needed.

This is a well-performed study. The manuscript is well-written with comprehensive presentation of the data.

Minor comments

The ICTV recommends using the abbreviation MCPyV for Merkel cell polyomavirus (MCV is also used as abbreviation for molluscum contagiosum virus, a poxvirus that was identified more than 50 years before MCPyV).

Figure 1A: How many parallels? SD?

Figure 2: the color codes of the small squares representing miR-375 OE and pcDNA3 are difficult to distinguish.

Figure 2A and 5A, right panel: Y-axis is different from left panel A in Figure 2A and from Fig1A and 3A

Figure 3A. The color code of the small squares representing miRNA mimic NC and miR-375 OE are hard to distinguish.

Line 124: define subtle (how many %?).

Figure 4C: MCC26 cell line: cleaved PARP is also observed in miRNA mimic NC.

Figure 4C: How many hrs after transfection was PARP cleavage monitored? How was PARP cleavage in stable miR-375 OE-transfected MCC14/2 cells?  

Figure 5A: The color code of the small squares representing pcDNA and miR375sp are difficult to distinguish.

Figure 6: The different color codes of the small squares are difficult to distinguish and does not seem to correspond with the color of the bars.

Line 186-187: starting with “Here, we demonstrated…”. Something seems to be missing in this sentence.

Line 319-320 : LDHB as therapeutic target is only mentioned for MCPyV-negative MCC patients, while conclusion of the abstract (lines 32-34) includes also virus-positive MCC patients.

Author Response

Point 1: “The ICTV recommends using the abbreviation MCPyV for Merkel cell polyomavirus (MCV is also used as abbreviation for molluscum contagiosum virus, a poxvirus that was identified more than 50 years before MCPyV). ”

Response: We thank the reviewer for this comment, and we have now changed the abbreviation of MCV to MCPyV throughout the text.

Point 2: Figure 1A: How many parallels? SD?” 

Response: We have now included four replicates with standard error mean in the revised Fig. 1A. The error bars are now stated in the figure legend.  

Point 3: Figure 2: the color codes of the small squares representing miR-375 OE and pcDNA3 are difficult to distinguish.  

Response: We have now replaced the graphs from patterns to color-codes. Furthermore, to improve the clarity of graphs presented throughout in the manuscript and considering the comments from Reviewer #1 we have similarly changed the graphs to color-coding in Figures 3, 5 and 6.  

Point 4: Figure 2A and 5A, right panel: Y-axis is different from left panel A in Figure 2A and from Fig1A and 3A "

Response: The endogenous miR-375 levels in all three MCPyV- cell lines were very low. Ectopic expression of miR-375 increased its levels many folds higher than the vector control. On the other hand, the endogenous miR-375 levels in MCPyV+ cell lines were much higher than in the MCPyV- cell lines and silencing of miR-375 reduced its levels to ~50% of the levels in the vector control. The high variation of endogenous (Fig. 1A) and exogenous (Fig. 2A, 3A and 5A) miR-375 levels in MCPyV+ and MCPyV- cell lines required application of different scales for the Y-axis. 

Point 5: Figure 3A. The color code of the small squares representing miRNA mimic NC and miR-375 OE are hard to distinguish."  

Response: We have now changed the graphs to color-coding.  

Point 6:  Line 124: define subtle (how many %?).” 

Response: As suggested by the reviewer, we have now added the percentage of increase (7-14%) in the text (line 170).  

Point 7:  Figure 4C: MCC26 cell line: cleaved PARP is also observed in miRNA mimic NC.”   

Response: MCC26 is the fastest growing cells among the three MCPyV- cell lines. The cleaved PARP observed in these cells could be due to over confluence in the MCC26 cells transfected with miRNA mimic NC and cell death occurred before harvesting for Western blot analysis. Alternatively, the effect could be due to the toxicity from transfection.  

Point 8: Figure 4C: How many hrs after transfection was PARP cleavage monitored? How was PARP cleavage in stable miR-375 OE-transfected MCC14/2 cells?”  

Response: The detection of cleaved PARP was evaluated after 48 h of transfection. This information is given at the heading of Figure 4 legend, and in addition, which is now specified in Fig. 4C legend. We also observed increased of cleaved PARP in the stable miR-375 OE-transfected MCC14/2 cells. Given all the experiments described in Fig. 4 were performed using miR-375 mimic and miRNA mimic NC, we propose not to include the result from the miR-375 OE stable cells in this figure. 

Point 9: Figure 5A: The color code of the small squares representing pcDNA and miR375sp are difficult to distinguish.”  

Response: We have now changed the graphs to color-coding.  

Point 10: Figure 6: The different color codes of the small squares are difficult to distinguish and does not seem to correspond with the color of the bars.” 

Response: We have now changed the graphs to color-coding.  

Point 11: Line 186-187: starting with “Here, we demonstrated…”. Something seems to be missing in this sentence.”  

Response: To improve the clarity of the text we have now revised this statement as follows: “Here, we demonstrated that LDHB mRNA and protein levels were reduced following over-expression of miR-375 and increased after suppression of miR-375, supporting that LDHB is a target of miR-375 in MCC.”  (line 276-278).  

Point 12: “Line 319-320 : LDHB as therapeutic target is only mentioned for MCPyV-negative MCC patients, while conclusion of the abstract (lines 32-34) includes also virus-positive MCC patients.” 

Response: We have now changed the statement and made a general conclusion as ”Targeting LDHB could be a novel therapy for MCC” (line 471-472).  

Round 2

Reviewer 1 Report

The manuscript has been great improved but one minor revisions should be made.

1. The molecular weight of cleaved PARP in Fig. 6C by the antibody they used should be 89 KDa, not 24KDa. The authors should carefully check it. 

Author Response

In this manuscript, we used two different PARP antibodies: BD Pharmingen (cat. no. 556362) and Abcam (ab32064). In Figure 4C, the antibody from BD Pharmingen was used, which can recognize the full-length (116 kDa) and the 89 kDa PARP. In Figure 6C, we used the antibody from Abcam, which specifically recognizes the 25 kDa cleaved form of PARP. The description of the antibodies are described in the Method section and also in the Figure legend 4C and 6C. We hope this will clarify the confusion of the size differences between Figure 4C and 6C.